

# A Machine Learning Method for Estimating Atmospheric Trace Gas Concentration Baselines

Kirstin Gerrand[1, 2], Elena Fillola[1, 3], Alistair J. Manning[4, 1], Jgor Arduini[5], Paul B. Krummel[6], Chris R. Lunder[7], Jens Mühle[8], Simon O'Doherty[1], Sunyoung Park[9], Ronald G. Prinn[10], Stefan Reimann[11], Dickon Young[1], and Matthew Rigby[1]

[1]Atmospheric Chemistry Research Group, School of Chemistry, University of Bristol, Bristol, UK
[2]Earth Sciences New Zealand, Private Bag 14901, Kilbirnie, Wellington, New Zealand
[3]School of Engineering Mathematics, University of Bristol, Bristol, UK
[4]Hadley Centre, Met Office, Exeter, UK
[5]Department of Pure and Applied Sciences, University of Urbino, Urbino, 61029, Italy
[6]CSIRO Environment, Aspendale, VIC, Australia
[7]NILU, Kjeller, Norway
[8]Scripps Institution of Oceanography, University of California San Diego, La Jolla, CA, USA
[9]Kyungpook Institute of Oceanography, Kyungpook National University, Daegu, Republic of Korea
[10]Center for Sustainability Science and Strategy, Massachusetts Institute of Technology, Cambridge, MA, USA
[11]Laboratory for Air Pollution/Environmental Technology, Empa, Dübendorf, Switzerland

**Corresponding authors:** Kirstin Gerrand (kigerrand@gmail.com) and Matthew Rigby (matt.rigby@bristol.ac.uk)

**Abstract.** Estimates of trace gas baseline mole fractions in high-frequency atmospheric measurement records are crucial for analysing long-term changes in atmospheric composition. Baseline mole fractions are those that would be observed far from
5 emission sources (and hence are representative of background conditions) at specific latitudes in the atmosphere. Previous methods for inferring baseline mole fractions have used statistical or meteorological approaches, or, if available, co-measured tracer species thought only to be emitted from non-baseline wind sectors. Combinations of these techniques have also been employed in some applications. Statistical methods typically fit a baseline to the observations themselves, while meteorological methods use atmospheric models of varying complexity to categorise air mass origins. In this paper, we present a novel machine
10 learning method for estimating trace gas baseline mole fractions, which benefits from the physical basis of model-based filtering without the need for running an expensive simulator. Our approach offers the accessibility and computational cost-effectiveness of statistical models, without the associated smoothing or difficulty in identifying rapid baseline variations. By training on historical Lagrangian particle dispersion model outputs, our model learns to predict baseline mole fractions directly from meteorological fields. This advancement opens new avenues for low-latency trace gas time series data analysis, reconstruction
15 of historical baseline trends, and improved utilisation of tracer measurement air mass classification methods.



## 1   Introduction

The evaluation of long-term trends in the concentration of atmospheric trace species is important for understanding phenomena such as stratospheric ozone depletion and climate change. However, a challenge associated with the analysis of high-frequency (∼ hourly to daily) trace gas measurements is the separation of "baseline" concentrations from measurements strongly influenced by nearby sources. Here, we define baseline measurements as those representative of concentrations that would be observed far from emission sources at the same latitude as the measurement point. Such baseline time series have become essential for understanding hemispheric or global trends in greenhouse gases (for example, the "Keeling curve" for carbon dioxide (Keeling et al., 2005)), and ozone depleting substances (e.g., Laube et al., 2022; Liang et al., 2022).

Whether the trace gas mole fraction in a particular air mass is characteristic of the regional baseline or non-baseline conditions depends on the interplay of meteorology and fluxes. Advection from regional sources or sinks to a measurement site over timescales of days to weeks will lead to mole fractions that are above or below the baseline. Enhancements above baseline can also be observed under low wind speeds or planetary boundary layer heights (PBLH), when the measurements are particularly sensitive to local fluxes. Even when the influence of regional fluxes on a particular air mass is small, variations in mole fractions are observed due to long-range transport from latitudes or altitudes with very different baseline mole fractions to that of the measurement site (e.g., Arnold et al., 2018; Lunt et al., 2016).

Measurement networks such as the Advanced Global Atmospheric Gases Experiment (AGAGE) collect mole fraction data for numerous greenhouse gases (GHGs) and ozone-depleting substances (ODSs) at several locations around the world at approximately hourly frequency (Prinn et al., 2018). These monitoring stations observe baseline concentrations, overlaid with time-varying enhancements or depletion events for the reasons outlined above. Data filtering is therefore needed to estimate baselines, or categorise the data points that best reflect baseline concentrations.

As an example, Fig. 1 shows AGAGE measurements of the hydrofluorocarbon, HFC-134a ($C_2H_2F_4$), at Mace Head, Ireland and Gosan, Republic of Korea. Widespread use of this compound as a refrigerant has resulted in increasing global atmospheric abundances (Liang et al., 2022). The measurements at Mace Head, Ireland, are characterised as baseline when air originates from the Atlantic to the West, but these can be overlaid with enhancements as a result, primarily, of the advection of "polluted" air masses from European sources to the East (raw data are shown as grey lines in Fig. 1). At Gosan, enhancements are seen when air originates from a wider range of wind directions, due to surrounding sources from China to the West, the Korean peninsula to the North and Japan to the East. Furthermore, during the summer months, intrusions of southern-hemispheric air are frequently seen, associated with the observation of below-northern-hemispheric mole fractions. The investigator may or may not wish to include baseline mole fractions originating from latitudes that are very different to the measurement station, depending on the application. For example, when estimating the long-term mole fraction trend using observations from Mace Head, Ireland, air masses originating from the tropical Atlantic were removed in Manning et al. (2021), but a summertime baseline more characteristic of the Southern Hemisphere was included in the inverse modelling study using Gosan data in Arnold et al. (2018).





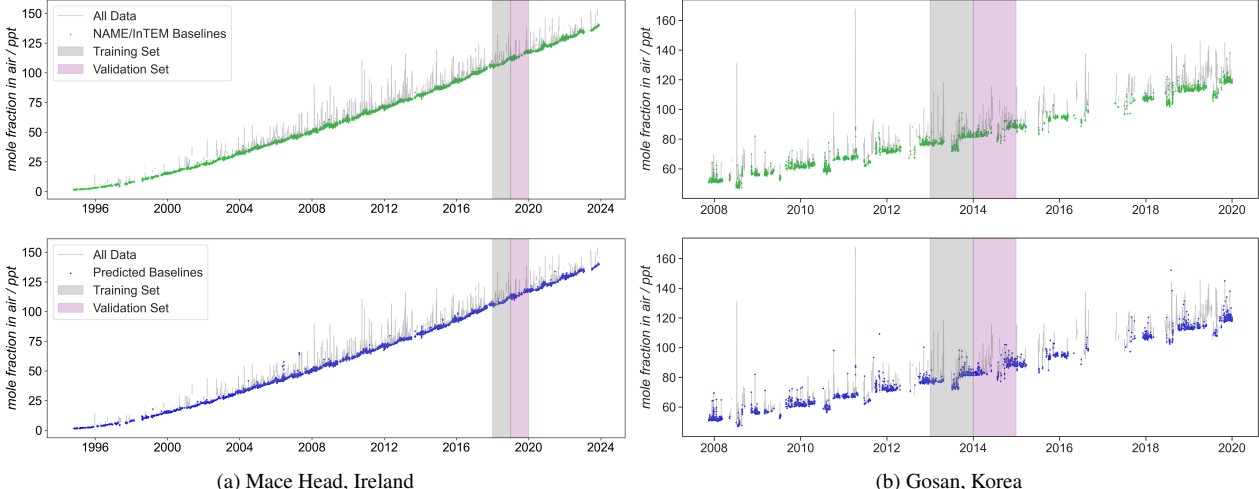

(a) Mace Head, Ireland  (b) Gosan, Korea

**Figure 1.** Measurement time series for HFC-134a ($CH_2FCF_3$) at a) Mace Head, Ireland and b) Gosan, Korea. The observations are shown in grey. The top panels show the measurements identified as baseline using the NAME/InTEM footprint-based filtering approach with no statistical filtering applied (green), and the bottom panels show the MLP algorithm's predictions of baseline points (blue). The one-year training period is shown as grey shading and the validation year is indicated by purple shading.

Previous baseline classification or fitting algorithms have broadly used three types of approach: statistical filtering, meteoro-
logical filtering or filtering based on co-measurement of some tracer for non-baseline conditions. Measurement-based methods have used species such as $^{222}$Rn or carbon monoxide (CO) to identify air masses substantially influenced by terrestrial sources or anthropogenic activity, respectively (e.g., Chambers et al., 2016, 2013; Yang et al., 2009). Since they require deployment of additional specialist instrumentation, we will not discuss them further here, although we note that our proposed approach could use measurements of these tracers as a training dataset.

Pure statistical filtering is seen in many studies, with examples including the use of iterative polynomial fitting with exclusion of outliers, and filtering of certain frequencies using Fourier transforms (O'Doherty et al., 2001; Ruckstuhl et al., 2012; Thoning et al., 1989; Novelli et al., 1998). O'Doherty et al. (2001) outlines in detail the AGAGE baseline algorithm, which iteratively fits a polynomial to the measurement time series and excludes points that are more than 3 standard deviations above the median within some time window (121 days). Similarly, Novelli et al. (1998) used a polynomial fit, including harmonic terms, with a
low-pass filter to exclude above-baseline measurements. Their approach was subsequently improved to transform the residuals to and from the frequency domain and apply high and low-pass filters in Novelli et al. (2003). Ruckstuhl et al. (2012) describe the "Robust Estimation of Baselines (REBS)" approach, which uses local regression within some defined time window to estimate the baseline and the distribution of its observed values.

Each of these statistical methods has the advantage of requiring no ancillary data or model simulations to apply, and is
computationally efficient. However, due to the use of polynomial fitting or moving windows over which data are excluded or regression is performed, they all implicitly or explicitly apply some smoothing to the data, which may not be characteristic





of the true baseline variability. Furthermore, they cannot readily identify or remove baseline values that are characteristic of a latitude different from that of the measurement station (as observed, for example, when southerly air masses arrive at Mace Head or Gosan; Fig. 1).

Meteorological filtering techniques have often used air mass back trajectories (estimates of advective transport prior to an observation), or wind sector analysis, to identify air masses that are unlikely to be strongly influenced by regional fluxes. Henne et al. (2008); Lööv et al. (2008); Salvador et al. (2010) computed back trajectories, and then applied a clustering algorithm to explore patterns in air mass origins. Alternatively, several studies (Derwent et al., 1998b; O'Doherty et al., 2001; Derwent et al., 1998a) have applied a wind sector allocation approach, described in detail by Derwent et al. (1998c) to isolate air masses

originating from some wind sector thought not to be strongly influenced by local fluxes.

In an extension of back-trajectory-based methods, recent studies have used trace gas source-receptor relationships ("footprints"), calculated using Lagrangian particle dispersion models (LPDMs), to estimate the full influence of transport and mixing on observed concentrations. A range of particle dispersion models have been applied to baseline estimation, including the UK Met Office Numerical Atmospheric Modelling Environment (NAME) (Ryall et al., 1998; Manning et al., 2011), which we use

in this study. LPDMs estimate footprints by considering the transport of an ensemble of hypothetical gas particles backwards in time, driven by archived meteorological fields. For the NAME simulations used in this study, these are obtained from the UK Met Office Unified Model analyses (Cullen, 1993). LPDM footprints quantify the contribution to the observed concentration of a unit emission from the surface at each grid cell surrounding the measurement point (Manning et al., 2011).

Manning et al. (2021) combined LPDM footprints with a population density map to identify air masses that were potentially

influenced by anthropogenic emissions (illustrated in Fig. 2). A flux proportional to population density was transported through the NAME model atmosphere to produce a synthetic anthropogenic tracer at each measurement site. When the concentration of this tracer fell below some arbitrary threshold, the corresponding air mass was labelled as being representative of the baseline. This method, part of the Inversion Technique for Emission Modelling (InTEM) (Manning et al., 2011), has been applied in a number of studies (Arnold et al., 2018; Lunt et al., 2021). In addition to baseline classification, the method also categorises

air masses further into a set of classes specific to a given site. For Mace Head, Ireland, for example, air masses are either "southerly" (when air masses originate from lower latitudes, specific for European sites), "local" (where local influences may dominate due to low ventilation conditions), "polluted" (where potential anthropogenic influence is high), "mixed" (when there is a combination of source types), "upper troposphere" (when air masses have descended from higher altitudes), or baseline. Figure 2 shows two example footprints for Mace Head, Ireland, superimposed on a population density plot to illustrate the

approach. Figure 2(a) shows conditions consistent with baseline mole fractions for compounds whose fluxes are highest over land; the footprint is primarily over the ocean and does not deviate substantially in latitude from the measurement location. Figure 2(b) shows an instance where the air mass originates from more populated and industrialised regions, where enhanced concentrations are typically observed.

Unlike statistical filters, meteorological (or model-based filtering) does not impose smoothing on the dataset and the baseline

classification can be justified based on physical considerations. However, the methods based on atmospheric model-derived air





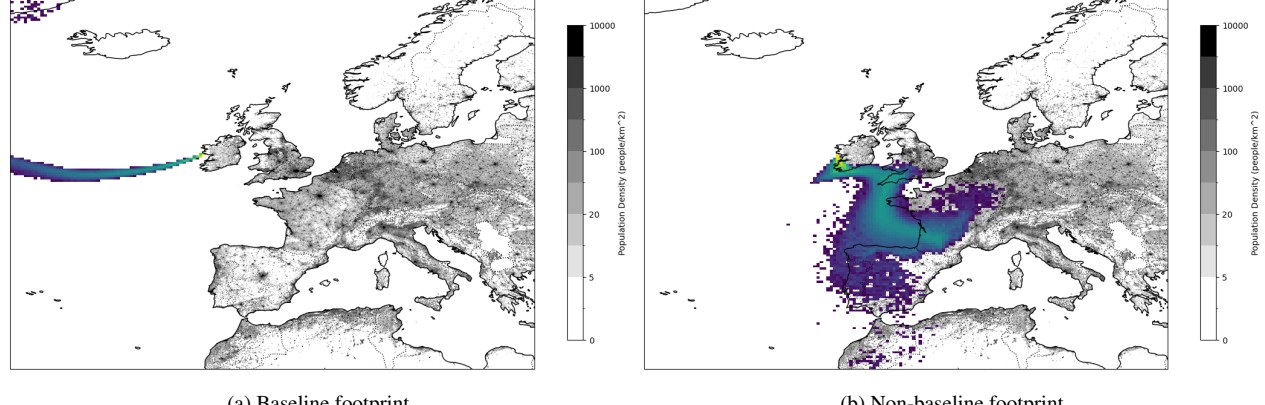

| (a) Baseline footprint | (b) Non-baseline footprint |

**Figure 2.** Two example NAME footprints showing incoming air masses to Mace Head atmospheric research station that illustrate meteorology consistent with, a) baseline and b) non-baseline observations. The footprint colour refers to the susceptibility of the measurement to emissions from that grid cell on a logarithmic scale; green represents higher values, and purple, lower (colourbar not shown). Population density is also shown, as a proxy for anthropogenic flux magnitude (grey shading).

histories are computationally expensive and rely on the availability of the appropriate meteorological datasets needed to drive models.

Here, we present a baseline classification algorithm using a machine learning (ML) approach that emulates an LPDM-based filter. This approach preserves the benefits of a meteorological filter for a fraction of the computational cost.

ML is a branch of artificial intelligence that enables computers to quickly analyse data and identify relationships and patterns within. It has been employed successfully for various applications in atmospheric chemistry, including the prediction of particulate matter concentrations, nitrogen dioxide modelling and estimating trace gas footprints (Brokamp et al., 2018; Masih, 2019; Fillola et al., 2023). Artificial neural networks, a type of supervised learning algorithm whose architecture mirrors the workings of the human nervous system and brain, have been demonstrated to be very useful in atmospheric science, especially

when dealing with complex systems with numerous non-linear relationships (Kruse et al., 2022; Gardner and Dorling, 1998; Lu et al., 2006). The multilayer perceptron (MLP) is one example of a widely used neural network architecture (Gardner and Dorling, 1998). Decisions are made by sending signals from an input layer to an output layer; data are processed in each step, with activation functions applied to introduce non-linearity to the computation (Gardner and Dorling, 1998). Another ML algorithm applied in atmospheric chemistry is the random forest; a collection of decision trees each make their own prediction

based on a unique subset of data, and the majority vote is the final prediction (Ivatt and Evans, 2020; Masih, 2019; Jiang and Riley, 2015).

The aim of this study is to investigate the application of ML for the classification of trace gas baseline air masses, in order to accurately recreate the air mass categories obtained from the Met Office NAME/InTEM LPDM-based algorithms. We demonstrate this method at nine different AGAGE sites, in a range of meteorological regimes and for a range of compounds.



The baseline classification is based only on meteorological inputs, namely wind speed and direction, boundary layer height
and surface pressure.

## 2    Methodology

### 2.1    Data

#### 2.1.1    Baseline Flags

For the purpose of training our ML algorithm, a dataset of flags that categorise air masses as "baseline" or "non-baseline"
using some independent method is required. Here, we use air mass labels, based on outputs from NAME/InTEM (Manning
et al., 2021; Jones et al., 2007). The method categorises air masses bi-hourly, with categories tailored for each site (e.g., "base-
line", "polluted", "local", "mixed", "southerly", "upper troposphere", for Mace Head, as described above). For this work,
the categories are simplified to a binary "baseline" versus "non-baseline" label (grouped non-baseline categories). In the In-
TEM framework, an additional statistical filter is applied to the derived baseline on a species-by-species basis, to remove any
remaining outliers, but the flags used in this study are based only on the footprint-based filter.

#### 2.1.2    Meteorological Data

Meteorological data were obtained from the European Centre for Medium-Range Weather Forecasting (ECMWF) ERA5 re-
analysis dataset (Hersbach et al., 2020). These data were downloaded at hourly intervals on a longitude/latitude horizontal
grid and hybrid coordinates in the vertical and used as an input to the ML algorithm. The ECMWF meteorological fields were
used, rather than the Met Office UM fields that were used to drive NAME, as they were more readily available for longer time
periods. An intercomparison of the two products showed close agreement at the AGAGE measurement stations, as would be
expected since both products assimilate similar meteorological observations. For example, calculated mean absolute percent-
age errors gave a less than 10 % error across a six-month sample period (January to June 2015) at Mace Head, Ireland, in the
wind direction. However, wind speed saw a slightly higher error (15.3 %) with UM speeds generally surpassing that of the
ERA-5 met products (see Supplementary Material).

#### 2.1.3    Mole Fraction Observations

Trace species mole fraction data were obtained from AGAGE (Prinn et al., 2025) at approximately two hour intervals. Mea-
surements are provided as dry air mole fractions (in ppt, equivalent to pmol mol$^{-1}$ or ppb, nmol mol$^{-1}$). We demonstrate our
algorithm at nine AGAGE sites, covering four continents and spanning a range of remote and relatively "polluted" environ-
ments. Details of the chosen sites are outlined in Table 1. All sites have at least 10 years of data, with most having more than
20. This is equivalent to over 200,000 measurements across the nine sites.



**Table 1.** Details of selected AGAGE sites, with site designation and coordinates given.

| Site Name | Station Designation | Coordinates |
|---|---|---|
| Kennaook/Cape Grim, Australia | CGO | 40.6833°S, 144.6894°E |
| Monte Cimone, Italy | CMN | 44.1932°N, 10.7014°E |
| Gosan, Korea | GSN | 33.2924°N, 126.1616°E |
| Jungfraujoch, Switzerland | JFJ | 46.5478°N, 7.9859°E |
| Mace Head, Ireland | MHD | 53.3267°N, 9.9046°W |
| Ragged Point, Barbados | RPB | 13.1651°N, 59.4321°W |
| Cape Matatula, American Samoa | SMO | 14.2474°S, 170.5644°W |
| Trinidad Head, United States | THD | 41.0541°N, 124.151°W |
| Zeppelin, Svalbard | ZEP | 78.9072°N, 11.8867°E |

## 2.2 Preprocessing

The mole fraction and meteorological datasets were combined by aligning them to the InTEM baseline flag time period. A
tolerance of one hour was set to avoid significant extrapolation of the mole fraction data.

Initial exploration of the datasets showed a significant imbalance between the baseline and non-baseline classes, with seven
of the nine sites seeing baseline points being less than one-third of the dataset. This was mitigated by undersampling the
majority class (non-baseline), which we found improved the predictive ability of the models. A random sample of non-baseline
points was removed, and we used a baseline:non-baseline ratio of 4:1 in our training set.

### 2.2.1 Model Inputs

The meteorological parameters included as inputs or features to the ML model were the eastward and northward wind compo-
nents ($\mathrm{m\,s^{-1}}$) at 10 meters above ground level and at two pressure levels, 850 and 500 hPa, surface pressure (Pa), and boundary
layer height (m). The wind data were interpolated to a 17-point grid system around each site, based on two 3x3 grids covering
$\pm$ 5 and $\pm$ 10 ° latitude and longitude. To provide the algorithm with further information on changes in wind speed and direc-
tion, u- and v-wind components were also provided for six hours prior to the measurement point. Additionally, two temporal
variables were added into the dataset to represent the time of day and the day of year. These two variables were integers ranging
from 0 to 23 and 1 to 366, respectively. This gave a total of 209 input variables. A list of all input variables can be found in the
Supplementary Material.

### 2.2.2 Dimensionality Reduction

To reduce the size of the input dataset and reduce the computational cost of model training and evaluation, a principle com-
ponent decomposition of the input dataset was used for one of the ML models we tested. As the inputs included a range of





physical quantities with different units and dynamic ranges, the data was first standardised by normalising by the mean and standard deviation per variable per atmospheric level (e.g., all eastward wind components at a given height). The first 20 principal components were used as ML model inputs from this standardised dataset. These components were found to explain over 80 % of the input data variance, whilst reducing the dimensionality of the input dataset by an order of magnitude.

### 2.3 Machine Learning Models

A multilayer perceptron (MLP) was trained to predict baseline classifications for each of the nine AGAGE sites. The method was also tested using a tree-based method (a random forest algorithm), the results of which can be found in the Supplementary Material. These methods outperformed alternatives, such as gradient boosted trees, in early testing, but it is likely that several other suitable architectures also exist.

The random forest method used the reduced dimension dataset. However, the MLPs performed best when provided with the full dataset. These choices were determined by balancing model evaluation metrics and training time.

#### 2.3.1 Model Training

Models were trained individually for each site. One year of data was used (approximately 1000 data points) for model training; 2018 was arbitrarily chosen for all sites except Gosan, which experienced substantial periods of down-time that year, so 2013 was used instead. The year following the training year was used for the validation set (2019 for all sites except 2014 for Gosan), and the rest of the dataset was used for testing. Given the substantial auto-correlation on the training dataset, characteristic of synoptic variability, it was important that the training and testing dataset be separated in time by greater than synoptic timescales (∼5-10 days).

Model hyperparameters were optimised using grid searches. These grid searches were performed independently for each site, but similar parameter choices were seen in the trained models throughout. For example, all MLP models used the ReLU (rectified linear unit) activation function, which outputs zero for all negative input values but leaves positive ones unchanged to introduce non-linearity to the dataset (Eckle and Schmidt-Hieber, 2019). All sites saw models with either four or five layers in their optimised MLP, with the number of hidden layers and the number of neurons they contain varying. The Mace Head model, for example, consists of an input layer with 209 neurons, two hidden layers with 200 and 150 neurons respectively, and a single-neuron output layer. Full hyperparameter sets can be found in the Supplementary Material.

A confidence threshold was introduced when making predictions, meaning that the model could only assign a baseline label when the associated confidence exceeded 80 %. This approach was introduced to improve the reliability of the model and reduce the occurrence of false positives.

#### 2.3.2 Model Evaluation

The model was trained to maximise F1 score (Eq.(1)), a measure of performance in binary classification problems that averages the precision and recall of a model (Eqs.(2) and(3)). Respectively, *precision* indicates the fraction of baselines identified




correctly, and *recall* identifies the fraction of predicted baselines that were "true" baselines. The model is evaluated through the use of these three metrics, as well as with a misclassification rate, which represents the proportion of instances in a dataset that
the model incorrectly classifies. The final values that are quoted for these metrics were calculated by considering all the data, except for those points used for training and validation. In most cases, the test set exceeded 20 years of data. Note that these metrics define the model's ability to separate baseline and non-baseline samples based only on the air mass meteorology, and so do not consider associated mole fraction values.

$$F1 = 2 \times \frac{(Precision \times Recall)}{(Precision + Recall)} \tag{1}$$

$$Precision = \frac{N(true\,positives)}{N(true\,positives) + N(false\,positives)} \tag{2}$$

$$Recall = \frac{N(true\,positives)}{N(true\,positives) + N(false\,negatives)} \tag{3}$$

To understand the performance of the ML model in application, the predicted labels can be applied to the atmospheric
measurements of a particular compound, and compared to the the NAME/InTEM baseline-only concentration time series. Measurement time series are often aggregated into monthly means in many applications, to remove the influence of short-term fluctuations (Laube et al., 2022; Liang et al., 2022). Therefore, the model was also evaluated by comparing the monthly means derived from the ML emulation to those from the InTEM baseline using three metrics; Mean Absolute Error (MAE), Root Mean Squared Error (RMSE) and Mean Absolute Percentage Error (MAPE). The monthly mean baseline was calculated by
collecting and taking the average of all mole fractions labelled as baseline in a calendar month.

### 2.3.3  Feature importance

We determine the variables that are most important for determining model performance by carrying out a permutation importance analysis (Breiman, 2001; Altmann et al., 2010). This approach works by randomly shuffling the values of a single feature and observing the change in the model's performance. This process is repeated multiple times to ensure robustness.
The importance of a feature is determined by the extent to which the model's performance degrades when the feature's values are permuted. The three most important features for each site are shown in the Supplementary material.

### 3  Results and Discussion

In this section we evaluate the MLP model by considering its ability to correctly identify baseline air masses based on meteorological inputs. We then compare time series for a selection of AGAGE gases. Summary results for all sites are presented,





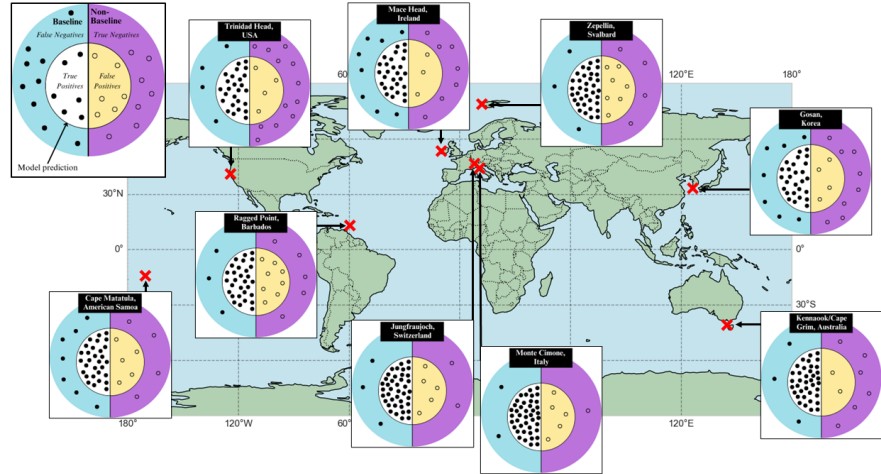

**Figure 3.** A map showing the locations of the nine AGAGE sites, with confusion matrix-derived plots at each location. Each confusion matrix was normalised, to reduce the visual impact of differences in testing set sizes; each point represents approximately 2 % of the total test set (rounding means that the total number of points on each plot range from 49 to 51). The left half of each circle represents true baseline points, and the right half true non-baseline points. The inner circle shows the MLP model prediction of baseline points. The key in the top left indicates where true or false positives and negatives lie, as explained in the main text.

and examples are shown for Mace Head, Ireland, and Gosan, Korea, chosen because of their very different local emissions and meteorological regimes (e.g., Fig. 1). Results for the random forest algorithm are presented in the Supplement.

## 3.1 Baseline Identification

Overall model performance for correctly classifying baseline versus non-baseline air masses was analysed by considering confusion matrices for each site. These matrices compare the predicted and "true" classifications of the testing set. Figure 3
shows the MLP confusion matrices for the nine AGAGE sites considered in this work. The outcomes of the confusion matrices have been normalised by testing set size and plotted as points on a precision/recall "bullseye" diagram. The inner circles are the MLP model predictions of baseline values. The left half of each diagram represents true baseline points, and the right-hand side are true non-baseline points. Therefore, the correctly identified baselines (true positives) lie in the left-hand segment of the inner circles (white). False positives, or points incorrectly identified as baseline, are in the yellow segment to the right
of the inner circle. Points in the blue segment (left-hand side of outer circle) represent missed baselines, or false negatives, whereas the purple area (right-hand side of the outer circle) are true negatives; points identified as non-baseline that are indeed non-baseline. A perfect model would only have points in the white area and the purple area. The fraction of points in the left half of the inner circle (white), compared to the total number in the inner circle (white and yellow), is the precision. The recall is the ratio of points in the inner left circle (white) to the total number of points on the left (white and blue).



**Table 2.** A tabular summary of the final MLP model outcomes, showing precision, recall and F1 score values for each of the AGAGE sites. Scores can range from 0 to 1, with higher scores indicating higher performance.

| Site Name | Station Designation | Precision | Recall | F1 score |
|---|---|---|---|---|
| Kennaook/Cape Grim, Australia | CGO | 0.942 | 0.930 | 0.936 |
| Monte Cimone, Italy | CMN | 0.897 | 0.962 | 0.928 |
| Gosan, Korea | GSN | 0.886 | 0.819 | 0.851 |
| Jungfraujoch, Switzerland | JFJ | 0.888 | 0.935 | 0.911 |
| Mace Head, Ireland | MHD | 0.906 | 0.794 | 0.846 |
| Ragged Point, Barbados | RPB | 0.811 | 0.916 | 0.860 |
| Cape Matatula, American Samoa | SMO | 0.866 | 0.822 | 0.844 |
| Trinidad Head, United States | THD | 0.878 | 0.874 | 0.876 |
| Zeppelin, Svalbard | ZEP | 0.844 | 0.928 | 0.884 |

The distribution of points in Fig. 3 reflects optimisation of F1 score in the MLP training, in which a high precision and recall is rewarded jointly; the majority of predicted baseline points are true positives (precision), and most of the available baseline points have been correctly identified (recall). These statistics are also summarised in Table 2. The table indicates that precision, recall and F1 score typically lie above 0.8. However, there is some variation between sites, with highest overall scores being obtained at Kennaook/Cape Grim, Australia, and poorer performance at Monte Cimone, Italy, which sees particularly

low recall (0.57). These differences are likely a product of the complexity of the meteorology at the site; Kennaook/Cape Grim is a coastal site, whose meteorology is dominated by strongly prevailing westerly flows, whereas Monte Cimone is a mountain site in a region of complex topography. Therefore, our features, which are a subset of meteorological variables on a relatively low-density grid around each site, may not capture well the flows in regions of more inhomogeneous topography and meteorology.

Our feature importance analysis (Section 2.3.3) reveals that wind components are typically the most critical for model performance, with PBLH generally being of lower importance (see Supplementary Material). However, it should be noted that these variables will be strongly correlated with one-another, so it is not possible to fully isolate the influence of each variable individually.

### 3.2   Baseline mole fractions

Whilst the above analysis demonstrates the model's ability to categorise air masses into baseline or non-baseline points, perhaps the most important test of the algorithm is in its ability to reproduce quantities that are used in real-world applications. Here, we focus on the simulation of baseline mole fractions and evaluate the model's ability to calculate robust baseline monthly means, which are commonly used to track changes in greenhouse gases and ozone depleting substances (e.g. Laube et al., 2022; Liang et al., 2022; Gulev et al., 2021).



**Table 3.** Atmospheric species used in this study. Atmospheric lifetimes are taken from Burkholder et al. (2023). Primary sources are from Laube et al. (2022); Liang et al. (2022). References describing the AGAGE mole fraction data are provided for some species in addition to Prinn et al. (2018, 2025).

| Species Name | Chemical Formula | Atmospheric lifetime (years) | Primary sources | Data citation |
|---|---|---|---|---|
| Methane | $CH_4$ | 12 | Wetlands, fossil fuels, agriculture, waste | Cunnold et al. (2002) |
| Nitrous Oxide | $N_2O$ | 109 | Agriculture | |
| CFC-12 | $CCl_2F_2$ | 102 | Refrigerants, aerosol propellants | Cunnold et al. (1983) |
| HCFC-22 | $CHClF_2$ | 12 | Refrigerants | Simmonds et al. (2017) |
| HFC-125 | $CHF_2CF_3$ | 31 | Refrigerants | Simmonds et al. (2017) |
| HFC-134a | $CH_2FCF_3$ | 14 | Refrigerants | Simmonds et al. (2017) |
| Dichloromethane | $CH_2Cl_2$ | 0.5 | Solvents, industrial processes | Simmonds et al. (2006) |
| Methyl bromide | $CH_3Br$ | 0.8 | Fumigants, industrial processes | |
| Sulfur hexafluoride | $SF_6$ | 850–1280 | Electrical insulation, magnesium production | Rigby et al. (2010) |
| Carbon tetrafluoride | $CF_4$ | 50,000 | Aluminium production, semiconductor manufacture | Simmonds et al. (1983) |

The model was applied to a subset of 10 atmospheric trace species measured by the AGAGE network (Prinn et al., 2025). These 10 compounds were chosen to span a range of sources, atmospheric lifetimes, and atmospheric histories (e.g., those that are growing in the atmosphere, such as HFCs, versus those that are declining, such as CFCs).

Examples of the baseline flags applied to HFC-134a measurements at Mace Head and Gosan are shown in Fig. 1. The figure reflects the accuracy of the categorisation shown in Fig. 3; the majority of the InTEM baseline points are identified

correctly. The influence of false positives is seen as a number of apparently above-baseline points that are categorised as baseline (noting that a small number of these elevated points appear to be incorrectly flagged as baseline in InTEM, which is why their full algorithm includes a subsequent statistical filtering step). At Gosan, similar to InTEM, the MLP algorithm identifies baseline values in the summer period, which sees Southern Hemispheric air intrusions. The rapid variation between northern and southern hemispheric baselines would be very difficult to detect with statistical filters, which rely on baselines

being smoothly varying.

When downsampled into monthly averages, the influence of false positives and false negatives is strongly muted. This is demonstrated in Fig 4, which shows that for almost all months, the error in the MLP model-calculated HFC-134a monthly mean at Mace Head and Gosan is substantially smaller than the baseline variability ($1\sigma$ standard deviation in the baseline) within a month. Where outliers occur, these tend to be during months where there were relatively few data points, meaning that

any mischaracterisation of the baseline can have a disproportionate influence.





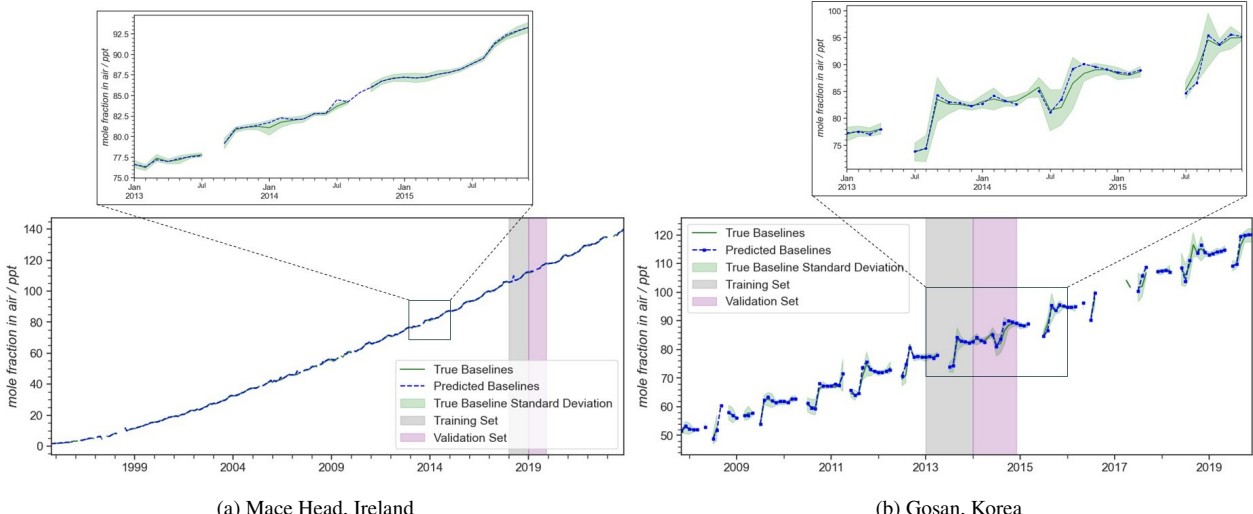

(a) Mace Head, Ireland

(b) Gosan, Korea

**Figure 4.** Baseline monthly means for HFC-134a at a) Mace Head, Ireland; b) Gosan, Korea. The NAME/InTEM baseline-derived values are in green, with associated $1\sigma$ variability within each month shown as green shading. The MLP model-predicted values are in blue. A subset of the dataset is shown in more detail in the top panels.

The overall performance of the algorithm applied to each species is summarised in Fig. 5, which shows the absolute error in the MLP-based monthly mean when compared to the InTEM-based equivalent, which is treated as the absolute truth here. The smallest values of MAPE appear to be associated with compounds that have relatively small gradients in the background atmosphere (e.g., CFC-12, $CF_4$, $N_2O$), which presumably reduces the impact of false positives or negatives. Likely for the opposite reason, shorter-lived compounds, such as dichloromethane, which tend to exhibit substantial seasonal variability and zonal gradients, tend to show the poorest model performance.

### 3.3 Computation and expected use cases

The computational cost of our baseline classification model is negligible compared to the calculation of LPDM footprints; on a standard desktop computer, the models consistently took less than a minute of wall time to train (1 year of data). Prediction of 1 year of baseline flags for ~hourly data, for the most part, takes less than 10 seconds. This should be compared to the calculation of LPDM footprints, which takes approximately 10 core minutes for each data point (the calculation of a year of baseline flags by combining the LPDM footprints and a tracer flux field is on the order of seconds to minutes of core time). For very long time series, such as those presented in Figs. 1 and 4, the main cost associated with our model is the retrieval of subsets of meteorological analyses, which can take a substantial time due to network and storage latency from some archival services (although, of course, substantially larger data volumes are required to run the LPDM).

The envisaged use-cases for the ML algorithm are partly related to its small computational cost, and partly related to its ability to categorise baselines without the need for LPDM simulations. For example, we envisage that this algorithm can be




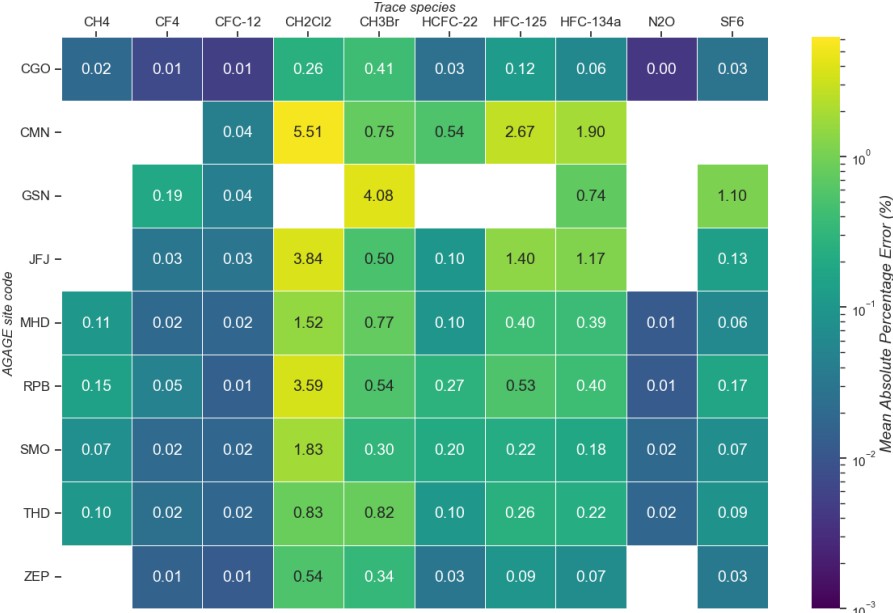

**Figure 5.** Mean Absolute Percentage Errors (MAPE) for the MLP model for selected AGAGE species across the nine monitoring sites. White denotes species that were not available at a particular site.

built into data visualisation and quality assurance software, so that mole fraction baseline trends can be examined by data owners with low latency. This is compared to the current situation, in which LPDM runs need to be performed and integrated

into the software, often by different research teams to those making the observation data, and often delayed by the availability of appropriate archived meteorology. These lags in the system can delay the analysis by weeks to months. Furthermore, our algorithm allows us to calculate baseline trends further back in time than is currently possible; at present, we only have NAME footprints from 1998 onwards, and it would require a major resourcing effort (both in terms of staff time and computation) to extend these model runs to the beginning of the AGAGE record (1978). Datasets such as ERA5 cover this period and earlier,

and can be used in the model prediction step. Finally, we envisage that this algorithm can be used to extend tracer-based methods beyond the period during which measurements were made. Consider, for example, a campaign in which $^{222}$Rn was used to derive baseline flags for one year at some location. Our algorithm can use these measurement-based flags in place of the NAME runs, and then be used to provide consistent flagging indefinitely beyond the study period.

These expected benefits must be weighed against the limitations of the model. Whilst we anticipate that improvements in

the model architecture or training will be possible, it will always be an approximation of the training data, which is itself, in this case, model-based. Therefore, categorisation errors occur in the form of false positives (points labelled as baseline that are not), or false negatives (baselines that are missed), and so the algorithm should not be used if a precise categorisation is needed of a small number of samples. However, if aggregating over multiple measurements, for example, when calculating a monthly mean of ∼ hourly data, we have shown that the model has high skill in the majority of cases.



# 4 Conclusions

We have presented a neural network-based model for baseline air mass classification, based only on meteorological data. The model preserves many of the benefits of LPDM footprint-based classification methods, but at a fraction of the computational cost. Precision and recall values of around 0.8 or higher were achieved for most sites examined here, and baseline monthly means were retrieved with uncertainties that were, in most cases, substantially smaller than baseline variability. We propose that this model can add value in low-latency data analysis and for extending baseline categorisation to time periods for which model simulations or co-measured tracers are not available.

*Code and data availability.* The code for the MLP models, trained models, data processing and evaluation tools are available at https://github.com/kgerrand/ml_baselines and https://doi.org/10.5281/zenodo.16923130. Training data (InTEM flags and extracted meteorology) are available at https://doi.org/10.5281/zenodo.16915853. Version 20250123 of the AGAGE data was used in this study (Prinn et al., 2025).

*Author contributions.* KG, EF and MR designed the research. KG performed the research under the supervision of EF and MR. AJM provided InTEM baseline flags. All other authors provided observation data, and all authors contributed to writing the manuscript.

*Competing interests.* The authors declare no competing interests.

*Acknowledgements.* The authors are grateful to the AGAGE team for their dedication in providing long-term records of high-precision, high-frequency observations. KG and MR were funded by the Natural Environment Research Council (NERC) InHALE Highlight Topic (Investigating HALocarbon impacts on the global Environment, NE/X00452X/1). MR was also funded by the UK Research and Innovation-funded projects Self-learning Digital Twins for Sustainable Land Management (EP/Y00597X/1) and Greenhouse gas Emissions Measurement and Modelling Advancement (GEMMA, NE/Y001761/1). EF was supported by a Google Research PhD Scholarship. We thank the NASA Upper Atmosphere Research Program for its continuing multi-decadal support of AGAGE, including full support of THD and SMO, and partial support of MHD, RPB and CGO stations, through grants 80NSSC21K1369 to MIT and 80NSSC21K1210 and 80NSSC21K1201 to SIO and earlier grants. The Department for Energy Security and Net Zero (DESNZ) in the United Kingdom supported the University of Bristol for operations at Mace Head, Ireland (contracts 1028/06/2015, 1537/06/2018, 5488/11/2021, and PRJ_1604) and through the NASA award to MIT with the sub-award to University of Bristol for Mace Head and Barbados (80NSSC21K1369). The National Oceanic and Atmospheric Administration (NOAA) in the United States supported the University of Bristol for operations at Ragged Point, Barbados (contracts 1305M319CNRMJ0028, and 1305M324P0411). Observations at Gosan, South Korea, and S.P. are supported by the Korea Meteorological Administration Research and Development Program (Grant No. RS-2025-02313790). Measurements at Jungfraujoch are supported by the Swiss National Programs HALCLIM and CLIMGAS (Swiss Federal Office for the Environment, FOEN), by the International Foundation





High Altitude Research Stations Jungfraujoch and Gornergrat (HFSJG), and by the European infrastructure projects ICOS and ACTRIS. Measurements at Zeppelin are supported by the Norwegian Environment Agency.



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
