# Peer review of "A Machine Learning Method for Estimating Atmospheric Trace Gas Concentration Baselines"

_EGUsphere, 2025_

## Referee Comment (RC1)

This manuscript presents a machine learning based surrogate framework for classifying baseline and non-baseline measurements at atmospheric monitoring sites. The study evaluates model performance across multiple global sites, chemical species, and multi-decadal time periods. It demonstrates that the proposed approach can reproduce key characteristics of baseline classification. This manuscript is generally well structured, and clearly motivated by practical limitations of computational bottlenecks in baseline classification.

Overall, this work represents a useful and timely methodological contribution with potential value for operational baseline filtering and retrospective analysis. However, several conceptual and interpretational aspects require clarification, particularly regarding the domain of applicability of the approach, optimization of the input features, and justifications about key aspects of the model development. Addressing the points outlined below would strengthen the manuscript and improve confidence in the broader use of the proposed method.

1. While the motivation for reducing the computational cost of LPDM-based baseline classification is clear, the manuscript lacks explicit justification of why a ML surrogate is required, as opposed to simpler statistical or reduced-order physical approximations. In particular, quantifying the computational savings, and discussing alternative non-ML approaches would strengthen the motivation for the proposed methodology.

2. A measurement is a combination of both background and enhancements from local sources, and this proposed approach performs a binary classification to identify measurements dominated by either background or emissions from local sources. This approach functions as a filtering rather than a decomposition method which can identify their contributions to the given measurement. I think this manuscript will benefit from a clarification on this distinction as well as a reasoning of why authors chose classification approach over decomposition. This will definitely help avoid over-interpretation of the resulting baseline time series.

3. Line 20: Why the baseline is defined as representative of concentrations far from sources at the same latitude as the measurement site? Shouldn't it depend on the meteorology? For example, if winds are coming from north/south then baseline will not be the representative from the same latitude.

4. Line 125: Why binary classification and not multiclass classification to classify various categories that authors mentioned for baseline and non-baseline cases?

5. Line 154: The training dataset seems to preserve the natural imbalance between baseline and non-baseline classes with 4:1 representation ratio. Many ML practitioners use balanced training set (1:1) even in cases where one class has more representation over another (e.g. fraud detection). The manuscript does not provide a justification for this choice or discuss its implications on model performance. It would be helpful for the authors to provide a justification after comparing these two approaches.

6. Line 159: Depending on the spatial domain, the air parcels may have a variable time when they entered the domain. It seems only using meteorology up to 6 hours before the measurement may not be enough, especially for large domains when the air parcels may have entered much earlier than 6 hours before measurements (e.g. 2-3 days earlier). This may also be an indication of overfitting if the performance is higher with just meteorology from 6 hours before measurement. The authors should add more meteorology data prior to measurement to strengthen the confidence in the physical representativeness of the approach.

7. Line 179: Why did the authors decide to train the model on only 1 year and test it on 20+ years?

8. Line 185: The authors trained separate models for each site instead of a single model. While this is a common approach in the field to train separate models for different sites, it would be interesting to discuss if a generalized model can be developed which can identify baseline across sites.

9. Line 221 and Table S5: I think it will also be interesting to analyze the feature importance from temporal perspective. For example, which timestep (measurement time or before) shows stronger feature importance.

10. Line 250: The observed feature importance patterns are consistent with recent work on ML-based emulation of LPDM footprints (e.g. FootNet). Citing and briefly discussing these related methods would help place the present results in the context of ML emulations studies.

11. Figure 1 & 4: It would help to explicitly clarify in both the figure captions that this ML model only classifies a measurement as a baseline or non-baseline rather than predicting the baseline mole fractions. As currently presented, Figures 1 and 4 could be interpreted as showing ML-predicted baseline mole fractions, whereas the concentrations shown seem to be the original observations filtered using the ML classification. Clarifying this would help avoid potential over-interpretation of the results.